# Integration of mRNA and miRNA Analysis Reveals the Post-Transcriptional Regulation of Salt Stress Response in *Hemerocallis fulva*

**DOI:** 10.3390/ijms24087290

**Published:** 2023-04-14

**Authors:** Bo Zhou, Xiang Gao, Fei Zhao

**Affiliations:** 1Key Laboratory of Saline-Alkali Vegetation Ecology Restoration, Northeast Forestry University, Ministry of Education, Harbin 150040, China; 2College of Life Science, Northeast Forestry University, Harbin 150040, China; 3Key Laboratory of Molecular Epigenetics of MOE, Institute of Genetics & Cytology, Northeast Normal University, Changchun 130024, China; 4Horticulture Science and Engineering, Shandong Agricultural University, Taian 271018, China

**Keywords:** salt stress, miRNA, *Hemerocallis fulva*, degradome, posttranscriptional regulation, differentially expressed gene, transcriptome, NaCl treatment

## Abstract

MicroRNAs (miRNAs) belong to non-coding small RNAs which have been shown to take a regulatory function at the posttranscriptional level in plant growth development and response to abiotic stress. *Hemerocallis fulva* is an herbaceous perennial plant with fleshy roots, wide distribution, and strong adaptability. However, salt stress is one of the most serious abiotic stresses to limit the growth and production of *Hemerocallis fulva*. To identify the miRNAs and their targets involved in the salt stress resistance, the salt-tolerant *H. fulva* with and without NaCl treatment were used as materials, and the expression differences of miRNAs–mRNAs related to salt-tolerance were explored and the cleavage sites between miRNAs and targets were also identified by using degradome sequencing technology. In this study, twenty and three significantly differential expression miRNAs (*p*-value < 0.05) were identified in the roots and leaves of *H. fulva* separately. Additionally, 12,691 and 1538 differentially expressed genes (DEGs) were also obtained, respectively, in roots and leaves. Moreover, 222 target genes of 61 family miRNAs were validated by degradome sequencing. Among the DE miRNAs, 29 pairs of miRNA targets displayed negatively correlated expression profiles. The qRT-PCR results also showed that the trends of miRNA and DEG expression were consistent with those of RNA-seq. A gene ontology (GO) enrichment analysis of these targets revealed that the calcium ion pathway, oxidative defense response, microtubule cytoskeleton organization, and DNA binding transcription factor responded to NaCl stress. Five miRNAs, miR156, miR160, miR393, miR166, and miR396, and several hub genes, squamosa promoter-binding-like protein (SPL), auxin response factor 12 (ARF), transport inhibitor response 1-like protein (TIR1), calmodulin-like proteins (CML), and growth-regulating factor 4 (GRF4), might play central roles in the regulation of NaCl-responsive genes. These results indicate that non-coding small RNAs and their target genes that are related to phytohormone signaling, Ca^2+^ signaling, and oxidative defense signaling pathways are involved in *H. fulva’s* response to NaCl stress.

## 1. Introduction

Environmental stress, including heat, drought, salinity, and low temperature, affects plant growth and development [1]. Salt stress is one of the major factors restricting agricultural production worldwide which can lead to the excessive uptake of ions (particularly Na^+^ and Cl^−^), a reduction in water uptake, a decrease in photosynthesis rate, the accumulation of reactive oxygen species (ROS), damage to cellular membranes, and various forms of physiological damage that affect plant growth and crop yield [2]. However, some plants have evolved sophisticated mechanisms, such as osmotic adjustment, ion homeostasis, hormonal regulation, antioxidant defense system, etc., to adapt to various abiotic injuries during their life cycle [3,4,5]. Plant salt tolerance depends on the ability to regulate gene expression to control physiology, metabolism, and growth. Besides transcriptional regulation, post-transcription regulation is also popular in plants to control gene expression during salt stress induction.

Plant microRNAs (miRNAs) are about 19 to 25 nucleotides of small non-coding RNAs and play key roles in plant development, signal transduction, metabolism, and responses to environmental stress. They regulate gene expression and RNA silencing via base pairing to complementary sequences with target mRNA at the post-transcription level [6]. Recent research has revealed the regulatory roles of plant miRNAs and their targets under salt stress. In cotton, 72 targets of 25 miRNAs with significant differential expression under salt stress are identified by degradome analysis and, among them, overexpression of mir1 (a novel miRNA identified in cotton), miR390, and miR393 can increase sensitivity to salt stress [7]. In *Fraxinus velutina* Torr, miR164d-NAC1, miR171b/c-SCL22, miR396a-GRF1/TGA2.3, miR160g-ARF18, miR156a/b-SBP14, miR8175-GR-RBP, miR319a/d-TCP2/4, and miR393a-TIR1 have also been identified in leaves and roots to be involved in salt stress responses [8]. Moreover, the miR156/SPL module enhances salt tolerance by upregulating *MdWRKY100* in apples [9]. Additionally, through overexpressing *Osa-MIR319b* and a target mimicry form of miR319 (*MIM319*), miR319 has been identified to target *PvPCF5* and positively regulate ethylene synthesis and salt tolerance in switchgrass [10]. Furthermore, in *Arabidopsis*, miR393-TIR1/AFB2 are regulated by Receptor for Activated C Kinase 1 (RACK1), which is a negative regulator of plant stress hormone-abscisic acid (ABA)-mediated pathways, and the down-regulation of auxin signaling through RACK1A-induced miR393 biogenesis potentially regulates the *Arabidopsis* acclimation to salinity [11]. Therefore, miRNAs induced in salt stress may target different pathway genes to regulate plant tolerance to abiotic stress through a complex network.

*Hemerocallis fulva* (day lily) is a very popular ornamental plant that is widely used in family gardening, landscaping, and other fields. Although the roots of *Hemerocallis fulva* have a history of being used medicinally in Asia, the species *H. citrina* (night lily, or long yellow daylily) produces popular vegetables and important medicinal plants with a long cultivation history. According to the research of Li et al., the night lily is likely the result of human selection from daylilies for human consumption during crop evolution [12] and Huang et al. have revealed neither colchicine nor its precursors in *H. citrina*, challenging the belief that this alkaloid causes poisoning by the plant [13]. By contrast, daylilies (*H. fulva*) can cause acute kidney injury if ingested by a cat [14]. *H. fulva* is widely adapted to plant growth and development on the seashore saline land and has a strong resistance to salt stress and drought, as well as high- and low-temperature stress. High-concentration salt stress has also been reported to significantly inhibit the photosynthetic capacity of *H. fulva* [15]. Currently, the regulatory roles of many miRNAs and target regulatory factors in plant development and abiotic stress response have been summarized and reviewed [16,17]. Furthermore, some miRNAs involved in the cold stress response of *H. fulva* have been identified [18,19]. However, the miRNAs and target genes involved in the salt stress tolerance of *H. fulva* need to be further explored. In this study, we identified the differential expression miRNAs–mRNAs related to NaCl-tolerance in *H. fulva*. Meanwhile, we performed an integration of miRNA and degradome sequencing analysis to identify the salt-responsive miRNAs and their target genes in the leaves and roots of *H. fulva* with or without NaCl treatment. The key regulatory network of miRNAs and targets in *H. fulva* responding to salt stress obtained from the study can provide new insights to understand the molecular regulation mechanism of salt tolerance in *H. fulva*.

## 2. Results

### 2.1. Identification of Conserved and Novel miRNAs in Hemerocallis

Twelve small RNA libraries from leaf and root samples of *Hemerocallis* under NaCl treatment and normal conditions were constructed for high-throughput sequencing. These libraries yielded an average of 13.05 million and 14.27 million raw reads for the no treatment leaf control (NT_leaf) samples and the NaCl treatment leaf (NaCl_leaf) samples, separately, and 17.33 million and 14.39 million raw reads for the no treatment root control (Nt_root) samples and NaCl treatment root (NaCl_root) samples, respectively (Appendix A). After removing low-quality reads, repeats, mRNA, tRNAs, rRNAs, snoRNAs, and snRNAs, an average of 3.03 million and 3.40 million valid reads for Nt_leaf and NaCl_leaf samples, separately, and 5.69 million and 1.35 million valid reads for Nt_root and NaCl_root samples, respectively, with the length of 18–25 nt were generated (Appendix A).

To identify the conserved and novel miRNAs in *H. fulva*, the valid unique reads were aligned to the miRBase v.22.0 database and *H. citrina* genome. A total of 1088 expressed miRNAs including 668 conserved miRNAs and 420 novel miRNAs in the sequenced libraries were identified (Appendix A). Of 18–25 nt sRNAs, the majority of miRNAs showed 18 nt, 21 nt, 22 nt, and 24 nt long, and 21 nt miRNAs had the highest abundance (Appendix A). According to the sequence similarity analysis, the conserved miRNAs were classified into 43 miRNA families, and MIR159, MIR2916, MIR396, MIR168, MIR166, MIR167, MIR171, and MIR156 families exhibited relatively high abundance (Appendix A). Based on the predicting hairpin structures, 18 novel miRNA candidates with high expression levels (the number of reads is higher than the average copy of the data set) were also identified. Among these candidate novel miRNAs, PC-3p-182_17973 was the most abundant in the leaf (Appendix A).

### 2.2. Differential Analysis of Candidate miRNAs in Roots and Leaves under NaCl Treatment

Based on the normalized expression levels, the Pearson correlation analysis between the samples was analyzed. The results showed that the relation of NT_root1 and 2, NaCl_root1 and 3, NT_leaf1 and 2, and NaCl_leaf1 and 3 is high, with the correlation coefficient R-value ≥ 0.921 (Appendix A). There were 72 and 109 miRNAs specially expressed in leaf and root, respectively, and 141 miRNAs co-expressed in both tissues. In the leaf, 61 and 39 miRNAs were specifically expressed in the NaCl treatment and normal conditions, respectively. However, in the root, the number of miRNAs specially expressed in the NaCl treatment and normal conditions was 17 and 98, respectively (Figure 1). Moreover, three miRNAs significantly up-regulated and seventeen down-regulated (*p* < 0.05) in roots were obtained under NaCl-treated conditions compared with normal conditions whereas, in leaves, only two miRNAs with higher expression levels and one miRNA with a lower transcription level (*p* < 0.05) were identified under NaCl treatment compared with normal conditions (Figure 2). Among the differentially expressed sRNAs in leaves and roots, conserved miR395a, miR171c, miR396b, and miR319b were up-regulated with the treatment of NaCl, while the remaining miRNAs, including conserved miR393a, miR166, miR159-p3, miR396b-p3, miR5076-p3, and miR171g, were down-regulated after NaCl treatment. Some candidate novel miRNAs were also detected to differentially express in the NaCl treatment leaves and roots (Appendix A).

### 2.3. Identification of Differential NaCl-Responsive Genes by RNA-Seq Analysis

In all 12 sequencing transcriptomes, the base percentage of Q30 was >97.5%, and the average mapping rate of the 12 samples was 78.05% (Appendix A). In total, 63,105 transcripts were identified in the leaf and root of *H. fulva*, and 54,295 transcripts were annotated to be encoding genes by searching against the COG, GO, KEGG, KOG, Nr, and Swiss-Prot databases. After normalization with FPKM (expected number of fragments per kilobase of transcript sequence per million base pairs sequenced), the NaCl-stress responsive genes were analyzed and identified. A total of 12,691 DEGs (differentially expressed genes) were detected between the root of the NaCl-treated and untreated samples, which included 6196 up-regulated genes and 6495 down-regulated genes in response to NaCl stress. However, in the leaf, only 1305 up-regulated genes and 233 down-regulated genes were detected under NaCl stress (Appendix A). 

GO enrichment analysis of the differentially expressed genes showed that biological processes including oxidation-reduction process (GO:0055114), protein ubiquitination (GO:0016567), photosynthesis (GO:0015979), flavonoid biosynthetic process (GO:0009813), ethylene-activated signaling (GO:0009873), metal ion transport (GO:0030001), response to stress (GO:0006950), and response to osmotic stress (GO:0006970) were significantly enriched in leaf. Whereas, in the root, the regulation of transcription (GO:0006355), protein phosphorylation (GO:0006468), response to salt stress (GO:0009651), and response to auxin (GO:0009733) were significantly enriched in biological processes. Moreover, oxidoreductase activity (GO:0016491), transition metal ion binding (GO:0046914), transmembrane transporter activity (GO:0022857), DNA binding (GO:0003677) related genes in leaf and protein binding (GO:0005515), DNA binding (GO:0003677), DNA binding transcription factor (Go:0003700), kinase activity (GO:0016301), oxidoreductase activity (GO:0016709), and transmembrane receptor protein serine/threonine kinase activity (GO:0004675) related genes in root were significantly enriched in molecular functions. Furthermore, chloroplast (GO:0009507), cytosol (GO:0005829), membrane (GO:0016020), thylakoid (GO:0009579), apoplast (GO:0048046), intracellular (GO:0005622), and nucleosome (GO:0000786) in leaf and plasma membrane (GO:0005886), an integral component of membrane (GO:0016021), membrane (GO:0016020), Golgi apparatus (GO:0005794), extracellular region (GO:0005576), and plasmodesma (GO:0009506) in root were significantly enriched in cellular components (Appendix A).

Differentially expressed mRNAs were annotated to 20 and 24 pathways, separately, in leaf and root (*p*-value ≤ 0.05) from the Kyoto Encyclopedia of Genes and Genome pathway (Appendix A). The MAPK signaling pathway (ko04016) and Photosynthesis (ko00195) were significantly enriched with differentially expressed genes in the leaf. In the MAPK signaling pathway, the genes encoding phosphate transporter PHO1-3, transcription factor WRKYY2, calcium-binding protein CML29, and mitogen-activated protein kinase were up-regulated and transcription factor EAT1-LIKE was down-regulated. While in the photosynthesis pathway, ATP synthase CF0 subunit I, ATP synthase subunit delta, subunit b’, photosystem I reaction center subunit N, XI, VI-1, and chloroplast photosystem II 10 kDa protein were all up-regulated under NaCl stress. However, the Ribosome pathway (ko03010) and Plant hormone signal transduction pathway (ko04075) were significantly enriched in the root. In the Ribosome pathway, the differential expression genes encoding acid phosphatase 1-like, 60S ribosomal protein L3 isoform X1, and cytochrome c biogenesis FN were up-regulated and 60S ribosomal protein L35-like, 40S ribosomal protein S15a, and ribosomal protein L33 were down-regulated. Whereas in the MAPK signaling pathway, the genes encoding mitogen-activated protein kinase NPK1-like isoform X1, transcription factor ICE1, and putative WRKY transcription factor 14 were down-regulated and the probable WRKY transcription factor 31 isoform X1 and the VQ motif-containing protein 8 and TT8L were up-regulated. The details of significantly differential genes in these pathways are presented in Appendix A. 

### 2.4. Identification of Cleaved miRNA Targets by Degradome Sequencing

Through total degradome library sequencing, we obtained 7,410,926 unique sequences, and 58.3% of these unique reads could be mapped to the transcripts of *H. fulva*. The mapped unique sequences were further analyzed using the CleaveLand pipeline to identify sliced miRNA targets. A total of 222 transcripts were identified as miRNA targets and divided into five categories (0–4) with *p* < 0.05 [20]. Among these miRNA targets, 119 pairs of miRNAs and their targets belonged to category 0 with the most abundant sequence reads at the cleavage site (Appendix A). In addition, 192 target genes that were regulated by 41 conserved miRNA families and 30 target genes that were regulated by 20 novel miRNA families were identified. Moreover, most of the targets of conserved miRNAs are annotated to proteins involved in the Plant hormone signal transduction pathway (ko04075) such as miR156 targeted squamosa promoter-binding-like protein 12 (HHC018925), miR160 targeted auxin response factor 18-like isoform X1 (MSTRG.28508), miR167 targeted auxin response factor 12-like isoform X1 (HHC048333) and auxin response factor 17 isoform X1 (HHC045314), miR171 targeted scarecrow-like protein 6 (HHC020999), and miR393 targeted transport inhibitor response 1-like protein (MSTRG.24459). These specific miRNA cleavage sites on candidate targets are shown with the t-plot in Figure 3. Other targets of conserved miRNAs and novel miRNAs were diverse and included transcription factors, signal transduction factors, and other proteins involved in various biological processes (Appendix A).

### 2.5. Transcriptome Profiles of miRNAs and Their Target Genes under NaCl Stress

To determine the correlation between miRNAs and their target genes, we compared the expression profiles of miRNAs and their targets in the root and leaf under NaCl treatment. Combining the analysis of degradome, the expression levels of twenty-two miRNA::target pairs were negatively correlated in the root and seven miRNA::target pairs showed opposite expression patterns in the leaf (Appendix A). The identified miRNA::target pairs included many previously identified conserved miRNAs and targets, such as the SQUAMOSA promoter binding protein-like (SPL) genes for miR156, auxin response factor (ARF) genes for miR160 and miR167, and the growth-regulating factor (GRF) gene as the target of miR396. Moreover, novel targets were also identified, including the MADS-box protein gene for miR6027, polyphenol oxidase for miR528, ubiquitin-conjugating enzyme E2 gene for miR399, copper/zinc superoxide dismutase for miR398, and hypothetical protein AXF42 as the target of miR5077. Furthermore, the novel PC-5p-69667_35 miRNA with its target hypothetical protein CKAN, PC-5p-1494_1921 with its target hypothetical protein TanjilG, and PC-3p-11056_265 with its target sodium-coupled neutral amino acid transporter 6 were also identified (Table 1). 

The NaCl-responsive miRNAs and their targets showed different expression patterns between leaves and roots. Compared with the normal-grown samples, miR156a showed as up-regulated while its target HHC018925 (SPL12) was down-regulated in roots under the NaCl treatment. Conversely, in leaves, miR156a was down-regulated and SPL12 (HHC018925) was also down-regulated under NaCl stress. Similarly, the expression patterns of miR160b and HHC035929 pairs were similar to the miR156 and SPL pairs in roots and leaves under NaCl stress. However, miR167a and HHC048333 pairs (ARF) displayed negative expression regulation both in roots and leaves after NaCl treatment. However, miR172 was up-regulated in roots and leaves, and its targets HHC037242, HHC003638, and HHC008487 were down-regulated in roots and leaves under NaCl treatment. The transcription level of miR393a was up-regulated and its target HHC029515 (transport inhibitor response 1-like protein, TIR1) was up-regulated in roots, but, in leaves, both miR393a and TIR1 were shown to be a little down-regulated. Moreover, the expression of miR396a was up-regulated in leaves and roots and its targets HHC039058 and HHC039468(GRF) were shown to be down-regulated under NaCl stress. Furthermore, miR398 showed as up-regulated and its target HHC005458 (copper/zinc superoxide dismutase, SOD) was down-regulated in roots. In addition, miR528 was up-regulated and its targets, HHC030646 and HHC030652 (polyphenol oxidase, PPO), were down-regulated in roots. However, the expression of miR6027 was down-regulated and its target HHC039860 (MADS-box protein 5) showed a higher expression in roots of the NaCl-treated sample (Figure 4). 

### 2.6. qRT-PCR Validation of NaCl Stress Response-miRNAs and -Target Genes in H. fulva

To verify the sequencing results, four miRNAs (miR396b, miR393a, miR160b, miR156a) and corresponding target genes *HHC020074* (*Growth-Regulating Factor 4-like*, *GRF4*), *HHC029515* (*Transport Inhibitor Response 1-like*, *TIR1*), *HHC035929* (*Auxin Response Factor 18-like*, *ARF18*), and *HHC018925* (*Squamosa Promoter-binding-Like protein 12*, *SPL12*) were confirmed by real-time PCR analysis. The results showed that the expressions of the four detected miRNAs are all up-regulated in the roots under NaCl treatment which is consistent with the sequencing results. However, among these target genes, the expression of *TIR1*, *ARF18*, and *SPL12* had no obvious difference whereas the transcription level of *GRF4* was higher in the root of the NaCl-treated sample than in that of the no treatment sample in the root (Figure 5). Meanwhile, in the sequencing results, the expression of *SPL12* was down-regulated in the root under NaCl treatment. Additionally, the expression of miR396b was up-regulated while the expression of miR156a was down-regulated in the leaves of the NaCl-treated sample, but the expression of miR396b was down-regulated in the leaves of the sequencing results. Moreover, the transcription level of miR160b and miR393a had no obvious difference in the leaves after NaCl treatment while the expression of miR393a was down-regulated in the sequencing results. Correspondingly, the expression of *ARF18* and *TIR1* was down-regulated while the transcription level of *SPL12* and *GRF4* had no obvious difference in the leaves under NaCl treatment. However, the expression of *GRF4* was a little down-regulated and *SPL12* was up-regulated while *ARF18* and *TIR1* had no obvious difference in the NaCl-treated leaves of sequencing (Figure 6). The qRT-PCR analysis also indicated that the expression changes of the target genes were not completely negatively correlated with the miRNAs with different expressions. 

### 2.7. miRNA-Mediated Gene Regulatory Network in Response to NaCl Stress in H. fulva

To illustrate the regulatory roles of NaCl-responsive miRNAs-targets pairs in *H. fulva*, we analyzed the expression of miRNAs and targets, and the target binding and splicing sites through degradome analysis. In the stress response network, miR396, miR159, miR166, miR171, miR156, miR393, miR5076, and miR395 with novel candidate PC-5p-361_8563, PC-3p-4499_652 are involved in the regulation of target expression. Most of the targets are transcription factors and stress-related genes, including WRKY transcription factor SUSIBA2-like, TIR1, SPL12, calmodulin-like protein 3, hsp70-Hsp90 organizing protein 2, NAC domain-containing protein 68, transcription repressor OFP7, growth-regulating factor 4-like, ethylene-responsive transcription factor ERF071-like, transcription factor TGA2.2-like, and Serine/arginine-rich splicing factor 33. Many miRNA-target pairs in the regulatory network are involved in the pathway of growth and development, signal transduction, and stress response. The regulatory network indicated the splicing regulation roles of these miRNAs to their targets in NaCl stress response in *H. fulva*. This network contains some enzymes that regulate the metabolic process, such as cinnamoyl-CoA reductase 1, homeobox-leucine zipper protein HAT5, thiamine thiazole synthase, and L-ascorbate peroxidase, indicating that complex regulation exists in NaCl stress responses in *H. fulva*.

## 3. Discussion

### 3.1. Analysis of Conserved and Novel miRNAs and Their Targets in H. fulva

Plant miRNAs play significant roles in physiological metabolism, growth, and development by regulating the expression of functional targets. Although many studies have explored molecular mechanisms and gene function analysis in the flower development and environmental stress of daylily (*H. fulva*) [19,21,22,23], the regulation of miRNAs and their functional targets in *H. fulva* under NaCl stress is still unclear. Here, we performed a miRNA–mRNA integration analysis in the leaves and roots of *H. fulva* with or without NaCl treatment and identified several crucial genes and signaling pathways involved in the NaCl stress response of daylily. Among the conserved *MIR159*, *MIR2916*, *MIR396*, *MIR168*, *MIR166*, *MIR167*, *MIR171*, and *MIR156* genes, miR156-targeted *SPL* have been elucidated to primarily relate to abiotic stress resistance and plant development in multiple pathways. Heterologously co-expressing *Osa-MIR156b* and *Osa-MIR156c* in alfalfa repressed the expression of *MsSPL6*/*12*/*13* and increased the tolerance to drought and salt stress [24]. However, *MIR156a* overexpression in apples down-regulated the expression of *MdSPL13* which targeted the gene promoter of *MdWRKY100* and led to a low transcription level of *MdWRKY100* and a weakened salt resistance [9]. In addition, the miR159-mediated *MYB33* and *MYB101* transcript degradation is involved in the regulation of ABA signaling [25]. MYB33 could also promote the transcription of *ABI5* (*ABA INSENSITIVE5*) by directly binding to its promoter and functioning upstream of miR156 in *Arabidopsis* [26]. Moreover, ABA acted as an important secondary signaling molecule to activate kinase cascades and mediate gene expression during the salt stress response [1,27]. Moreover, the miR166-targeted *HD-ZIP* (*Homeodomain-leucine zipper*) [28,29], miR167-targeted *ARF* [30,31], miR168-targeted *AGO* (*Argonaute*) [32], miR171-targeted *SCL* (*Scarecrow-like*) [33,34], and miR396-targeted *GRF* (*Growth regulating factor*) [35,36] have also been identified to be involved in abiotic stress adaptation. Furthermore, miR156, miR159, miR167, miR168, miR171, miR319, and miR396 were also found to differentially express after salt stress in Radish [37], *Arabidopsis* [38], and *Zea mays* [39]. Through degradome analysis, we obtained the multi-targets of miR2916, such as *RNF14* (*Ring Finger Protein 14*, E3 ubiquitin-protein ligase), *SUD1* (*SUPPRESSOR OF DRY2 DEFECTS1*, E3 ubiquitin ligase), nematode resistance protein-like *HSPRO2*, *ornithine decarboxylase-like*, *NDR1/HIN1-like protein 13*, *phosphoglycerate kinase*, and *protein FAR1-RELATED SEQUENCE 5-like*, and some of them were reported to be involved in the resistance to biotic and abiotic stress [40,41,42]. Additionally, *dormancy-associated protein homolog 3-like*, the target of novel miRNA PC-3p-100300_22, and *elongation factor 1-alpha*, the target of novel miRNA PC-3p-25677_112, have also been identified to be related to stress and development response [43,44].

### 3.2. Differential Expression of the Identified miRNAs and Their Target Genes

One miRNA can target multiple genes to regulate the expression of these genes and one gene can also be simultaneously targeted by more than one miRNA. In this study, we identified 23 miRNAs’ differential expression in *H. fulva* under NaCl stress. Through degradome analysis, 222 targets were spliced by these miRNAs (Appendix A). Among the targets of the miRNAs regulated, 62 targets showed differential expression in the root and leaf of *H. fulva* under NaCl stress (Appendix A). Furthermore, the expression of different targets of one miRNA was up-regulated or down-regulated. For example, the expression of *heat shock protein 81-1* (rna-HHC004013.1, rna-HHC009402.2, and rna-HHC049019.1) targeted by aof-miR396b_L-1R-1 was down-regulated in the root of NaCl stress while the transcription levels of *acyl-CoA-binding domain-containing protein 3-like* (rna-HHC035500.1) and *ethylene-responsive transcription factor ERF071-like* (rna-HHC039932.1) were high in the root of NaCl stress. In addition, the *HSP81-1* (*HEAT SHOCK PROTEIN 81-1*) was reported to be down-regulated in salt-acclimated *Nicotiana tabacum* cells in comparison with control cells growing in the logarithmic phase [45]. The result indicated that the up-regulation of miR396 negatively regulates the expression of *HSP81-1* involved in the response of NaCl stress in the root of *H. fulva*. Moreover, in rice, *OsACBP4* and *OsACBP5* (acyl-CoA-binding domain-containing protein) mRNAs were induced after salt treatment and then remained at relatively high levels [46]. *AtERF71*/*HRE2*-overexpressing transgenic plants also showed tolerance to salt stress and other stress in *Arabidopsis* [47]. This suggested the negative regulatory role of miR396 to the *ACBP3* and *ERF071* might contribute to maintaining the relatively stable transcription level of *ACBP3* and *ERF071* under abiotic stress and to increase tolerance of *H. fulva* under NaCl stress. 

In the present study, we also found that the miR396b/GRF4 module, the miR166e/calmodulin-like protein 3 modules, and miR159/NAC domain-containing protein 68 display differential expressions in the root of NaCl stress. The miR396b/GRF module has been reported to be involved in abiotic stress responses in Pitaya (*Hylocereus polyrhizus*) [48]. The balanced opposing activities and physical interactions of the rice GRF4 transcription factor and the growth inhibitor DELLA conferred homeostatic co-regulation of growth and the metabolism of carbon and nitrogen [49]. Then, the miR396-GRF-GIF-SWI/SNF module played an essential role in GA signaling to control plant growth [50]. In addition, the miR166e/calmodulin-like protein module was newly identified in our research, and the miR166 has been revealed to be involved in salt stress response [51] and GA biosynthesis and catabolism [52]. Furthermore, calmodulin-like (CML) proteins, such as Ca^2+^ sensors, play important roles in the regulation of plant development and stress responses [53,54]. Moreover, miR159/NAC domain-containing protein 68 was also the new module identified through degradome in the root of *H. fulva*. The ClNAC68 positively regulated sugar and IAA accumulation in watermelon [55], and overexpressing *MusaNAC68* elevated the tolerance of salinity and drought stress in bananas [56]. In our research, the expression of *aof-MIR159-p3_1ss9TA* and *ptc-MIR166e-p5_2ss19TA21TC* was down-regulated and *aof-miR396b_L-1R-1* was up-regulated in the root of the NaCl-stressed sample. However, their corresponding targets showed an opposite expression trend in the root under NaCl stress. The results implied that miR396b/GRF4 module, miR166e/calmodulin-like protein 3 modules, and miR159/NAC domain-containing protein 68 are also involved in the response of NaCl stress in the root of *H. fulva*. 

With the qRT-PCR verification, we found the expression of *SPL12* is not consistent with the result of sequencing in the root. Further analysis of the result of sequencing showed that miR156a can target regulating multiple *SPL* genes. Both HHC018925 and HHC023313 were annotated to SPL12, but the expression of *HHC018925* is down-regulated while the expression of *HHC023313* is up-regulated in the root of the NaCl-treated sample. Multiple sequence alignment analyses indicated that HHC018925 and HHC023313 have high sequence similarity and the primers of qRT-PCR can specially bind both of them for amplification. The qRT-PCR detected expression of *SPL12* was the total transcription level of HHC018925 and HHC023313 (Figure 5). The different transcription level of different *SPL* gene members has also been identified in alfalfa (*Medicago sativa* L.) [24] and tamarisk (*Tamarix chinensis*) [57]. After 2.5 h NaCl treatments, the expression of *TcSPL1*,*2*,*3*,*4*,*5*,*7*,*8* was down-regulated while the expression of *TcSPL6*,*9*,*10*,*11*,*12* was up-regulated in *Tamarix chinensis* roots [57]. Then, different family gene members showed different expression characters in response to the same environmental treatment. For example, *MIR156a* overexpression weakened salt resistance in apples [9] while overexpressing *MIR156* in alfalfa showed improved salt and drought tolerance [24]. This could partially answer the same miRNA taking opposite regulating roles in the identical stress treatment of different plant species. 

### 3.3. miRNAs Regulate NaCl Stress Response Partially via Phytohormone Signaling, Ca^2+^ Signaling, and Oxidative Defense Signaling Pathways

Plants adapt to abiotic stress through miRNA-induced mechanisms at different signaling pathways. Plant hormones play a critical role in plant development and resistance to abiotic stress. In olive (*Olea europaea*) plants, low-salt pre-exposure primes against high salinity stress, and hormone signaling and defense-related interactions are reported to be associated with the priming responses [58]. Our results also showed that miR396b down-regulated *Growth-Regulating Factor 4-like* (*GRF4*, *HHC039058*) expression and up-regulated ethylene-responsive transcription factor *ERF71-like* (*HHC039932*) expression. GRF4 has been proven to be involved in the GA signaling pathway of plant development and abiotic stress response [49,59,60,61]. Moreover, OsmiR396c-OsGRF4-OsGIF1 regulatory module was reported to determine grain size and yield in rice and miR396b-GRF regulation also displayed abiotic stress responses in Pitaya (*Hylocereus polyrhizus*) [48,62]. Additionally, ethylene response factors (ERFs) are important ethylene-signaling regulators and are involved in abiotic stress response [47,63,64]. Furthermore, transport inhibitor response 1(TIR1) is an auxin receptor involved in the auxin-mediated signaling pathway regulating roots and hypocotyl growth and salt tolerance response [65,66]. In our research, the expression of *HHC014389* (*TIR1*) targeted by miR393a was also up-regulated in the root under NaCl stress. Auxin response factors (ARFs) have been confirmed to play crucial roles in plant developmental responses and abiotic stress by regulating the auxin signaling pathway [67,68]. A previous study has also shown that the overexpressed miR393-resistant form of *mTIR1* enhances salt tolerance by increased osmoregulation and Na^+^ exclusion in *Arabidopsis thaliana* [65]. These results indicate that miR393a/TIR1 module enhances the tolerance of *H. fulva* to salt stress via mediating the auxin signaling pathway. Together, miR396b/GRF4, miR396b/ERF71, and miR393a/TIR1 modules enhance tolerance to salt stress by regulating the GA, ethylene, and auxin signaling pathways in *H. fulva*.

Calmodulin (CaM) and calmodulin-like proteins (CMLs) are major Ca^2+^-binding proteins regulating the activity of a wide range of effector proteins in response to calcium signals. Ca^2+^-binding proteins sensed the stress signals and bound Ca^2+^ triggering their conformational changes to regulate downstream targets [54]. *CML9*, *CML24*, *CML37*, *CML38*, and *CML39* transcripts have been identified to be induced by salt and/or drought stress [69]. The overexpression of *calmodulin-like 44* (*ShCML44*) enhanced the tolerance of tomatoes in cold, drought, and salinity stresses [70]. In this study, we found that miR166e targeted *CML3* (HHC010057) and up-regulated the expression of *CML3* in the roots under salt stress, suggesting that miR166e/CML3 module increases salt resistance through regulating Ca^2+^ signal transduction. The increasing evidence has also shown that Ca^2+^/CaM-mediated signaling is involved in regulating ROS-related signal transduction [71]. Thus, the candidate miR166e/CML3 module contributes to the enhanced salt tolerance in *H. fulva*.

Multiple studies have proven that squamosa promoter-binding-like protein (SPL) can regulate salt tolerance in many plants, such as apple [9], alfalfa (*Medicago sativa* L.) [72], and tamarisk (*Tamarix chinensis*) [57]. Recent research showed that silencing CaSBP12 in pepper (*Capsicum annuum*) enhances tolerance to salt stress and reduces ROS accumulation [73]. In this study, we found that miR156a/c/g targeted *SPL1* (*HHC037962*), *SPL7* (*HHC016666*), and *SPL12* (*HHC018925* and *HHC023313*) and displayed different expression characters in the root and leaf of *H. fulva* under salt stress. The miR156-SPL12 module might increase ROS scavenging and lead to enhanced salt tolerance. Furthermore, APX (ascorbate peroxidase), which can detoxify ROS in plant abiotic stresses [74], has also been identified to be targeted by miR396b and up-regulated in root and leaf of *H. fulva* under salt stress in our study. Another gene, *HSP81* (*heat shock protein 81*), was also proved to be targeted by miR396b and down-regulated in root under salt stress. HSPs can function in enhancing membrane stability and detoxifying ROS in plant abiotic stress resistance [75]. However, different family members of *HSP* displayed up-regulated or down-regulated expression in salt-acclimated cells compared with the control cells of *Nicotiana tabacum* suggesting the complex regulation of HSP in plant salt stress tolerance [45]. Consequently, miR156 -SPL12, miR396b/APX, and miR396b/HSP modules are involved in regulating ROS scavenging and enhancing salt stress tolerance in *H. fulva*.

## 4. Materials and Methods

### 4.1. Plant Materials and Salt Stress Treatment

Two year-grown plants of ‘Chengsebaoshi’ (Appendix A), a cultivated variety of *H. fulva*, were planted on the campus of Northeast Forestry University, Harbin, China, (126°37′ E, 45°42′ N). The uniform-growth plants were potted in an experimental field and watered with 200 mM NaCl once a day (it has been reported that 250 mM NaCl can significantly inhibit the photosynthetic capacity of *H. fulva*, therefore 200 mM NaCl was used in this research) [15]. After two weeks of treatment, the roots and leaves of *H. fulva* were collected as treatment samples, and the plants without NaCl treatment were considered as the control. There were three biological replicates for each treatment. All the samples were collected, flash-frozen in liquid nitrogen, and used for total RNA extraction and high throughput sequencing. 

### 4.2. Small RNA, RNA-Seq, and Degradome Library Construction and Sequencing

A total of 12 samples (roots and leaves of *H. fulva*, and two treatments with and without NaCl treatment with three biological replicates) were used for small RNA, RNA-Seq, and degradome library construction and sequencing. Total RNA was isolated and purified using TRIzol reagent (Invitrogen, Carlsbad, CA, USA) following the manufacturer’s procedure. The RNA amount and purity of each sample were quantified using NanoDrop ND-1000 (NanoDrop, Wilmington, DE, USA). The RNA integrity was assessed by Agilent 2100 with an RIN number > 7.0. For small RNA sequencing, the library of each sample was constructed using TruSeq Small RNA Library Preparation Kit (Illumina, San Diego, CA, USA), according to the manufacturer’s protocol. For RNA-Seq, the cDNA library was prepared by purifying the RNA from total RNA using Dynabeads Oligo (dT)25-61005 (Thermo Fisher, Waltham, MA, USA). Next, the mRNA was fragmented into small pieces via Magnesium RNA Fragmentation Module (NEB, cat. E6150S, USA). Then, SuperScript™ II Reverse Transcriptase (Invitrogen, cat. 1896649, USA) was used to reverse-transcribe the cleavage RNA pieces according to the guidance for the mRNASeq Sample Preparation Kit RS-122-2103 (Illumina, San Diego, USA). After U-labeled second-stranded DNA synthesis and adding A-base to the blunt ends, the indexed adapters were ligated to the A-tailed fragmented DNA and the ligated products were amplified with PCR. The average insert size for the libraries was 300 bp (±50 bp). For degradome sequencing, the library from equally pooled 12 samples was constructed using poly-T oligo-attached magnetic beads purification and RNA ligase for the 5′ adapter ligation to the 5′ end of the 3′cleavage product of the mRNA. Then, the ligated RNA was reverse-transcribed to cDNA and amplified with PCR by the following conditions: initial denaturation at 95 °C for 3 min; 8 cycles of denaturation at 98 °C for 15 s, annealing at 60 °C for 15 s, and extension at 72 °C for 30 s; and then final extension at 72 °C for 5 min. Both small RNA and degradome libraries were sequenced on an Illumina Hiseq 2500 instrument (LC-Bio, Hangzhou, China) at Single-end (50 bp). The cDNA library was performed Pair-end (150 PE) on an Illumina Novaseq™ 6000 (LC-Bio Technology CO., Ltd., Hangzhou, China).

### 4.3. Identification of miRNAs

Raw reads were subjected to an in-house program, ACGT101-miR (LC Sciences, Houston, TX, USA), to remove adapter dimers, junk, low complexity, common RNA families (rRNA, tRNA, snRNA, snoRNA), and repeats. Subsequently, unique sequences with lengths in 18–25 nucleotides were analyzed by BLAST [76] against miRBase 22.0 (http://www.mirbase.org/, accessed on 13 October 2021) to identify known miRNAs and novel 3p- and 5p-derived miRNAs. Length variation at both 3′ and 5′ ends and one mismatch inside of the sequence were allowed in the alignment. The unique sequences mapping to mature miRNAs in hairpin arms were identified as known miRNAs and the sequences mapping to the other arm of known precursor hairpin opposite to the annotated mature miRNA-containing arm were considered to be novel 5p- or 3p-derived miRNA candidates. The unmapped sequences were analyzed by BLAST against the *Hemerocallis citrina* genomes [13], and the hairpin RNA structures containing sequences were predicated from the flank 120 nt sequences using RNAfold software (version 2.4.18) (http://rna.tbi.univie.ac.at/cgi-bin/RNAWebSuite/RNAfold.cgi, accessed on 13 October 2021). The potential miRNA precursors must meet the criteria as follows: (1) the number of nucleotides in one bulge in the stem (≤12); (2) the number of base pairs in the stem region of the predicted hairpin (≥16); (3) cutoff of free energy (kCal/mol ≤ −15); (4) length of the hairpin (up and down stems + terminal loop ≥50); (5) length of the hairpin loop (≤200); (6) number of nucleotides in one bulge in the mature region (≤4); (7) number of biased errors in one bulge in the mature region (≤2); (8) number of biased bulges in the mature region (≤2); (9) number of errors in the mature region (≤4); (10) number of base pairs in the mature region of the predicted hairpin (≥12); (11) percent of mature in stem (≥80).

### 4.4. RNA-Seq, Transcriptome Assembly, and Annotation

Cutadapt software (version cutadapt-1.9)(https://cutadapt.readthedocs.io/en/stable/, accessed on 13 October 2021) was used to remove the reads that contained primers and adaptor contamination and reads with low-quality bases. The clean reads were mapped to the *Hemerocallis citrina* genome by HISAT2 software (version 2.2.0) (https://daehwankimlab.github.io/hisat2/, accessed on 13 October 2021) [77]. Then, the mapped reads of each sample were assembled using StringTie (version 2.2.0) (http://ccb.jhu.edu/software/stringtie/, accessed on 13 October 2021) with default parameters [78]. All transcriptomes from all samples were merged to reconstruct a comprehensive transcriptome using gffcompare software (version 0.12.6) (http://ccb.jhu.edu/software/stringtie/gffcompare.shtml, accessed on 13 October 2021) [79]. StringTie (version 2.2.0) (https://ccb.jhu.edu/software/stringtie/) and ballgown (version 2.30.0) (http://www.bioconductor.org/packages/release/bioc/html/ballgown.html, accessed on 13 October 2021) were used to estimate the expression levels of all transcripts and perform expression level for mRNAs by calculating FPKM (FPKM = [total_exon_fragments/mapped_reads (millions) × exon_length (kB)]). The differentially expressed mRNAs were selected with fold change > 2 or fold change < 0.5 and *p*-value < 0.05 by R package edgeR (version 3.40.2) (https://bioconductor.org/packages/release/bioc/html/edgeR.html, accessed on 13 October 2021) [80]. The Benjamini–Hochberg (BH) algorithm was used to adjust the *p*-value for controlling the false discovery rate (FDR). Unigenes were then annotated to query against the Swiss-Prot, Non-redundant (Nr), Protein family (Pfam), Kyoto Encyclopedia of Genes and Genomes (KEGG), eukaryotic Orthologous Groups (KOG), Gene Ontology (GO) public databases by BLASTx with an E-value < 10^−5^. Finally, GO enrichment [81] and KEGG enrichment [82] analysis and the heatmaps of differentially expressed genes (DEGs) among four tissue samples were performed using the OmicStudio tools at https://www.omicstudio.cn/tool (accessed on 13 October 2021). 

### 4.5. Degradome Sequencing Data Analysis and miRNA Targets Identification

After degradome sequencing, the data were analyzed using the CleaveLand pipeline v.3.0 to identify cleaved miRNA targets [20]. The sequencing reads were mapped to the assembled *F. fulva* mRNA sequences, and the perfect matching alignments were extracted by using the Oligomap short reads aligner [83]. The 26 nt extracted mRNA sequences were subsequently aligned to the identified mature miRNAs by Needle program [84]. Then, alignments, where the degradome tag position coincided with the 10th or 11th nt of a given miRNA, were kept and scored [85]. Moreover, t-plots showing the distribution of signatures of miRNA-cleaved targets were built using R package.

### 4.6. Differential Expression Analysis of miRNAs–mRNAs in Different Tissues of H. fulva

The miRNA reads were normalized to total reads by transcripts per kilobase million, and the members from a given miRNA family with identical sequences were classified into a unique miRNA. The differential expression of miRNAs (DEmiRs) between each group of treated and untreated roots or leaves of *H. fulva* were identified using R package DEseq2 [86], with |log2 fold change| ≥ 1 and *p*-value ≤ 0.05. After log2 transformation and normalization, the candidate miRNA–mRNA pairs containing DEmiRs and differentially expressed genes (DEGs) were selected as miRNA-target pairs, and, for each selected pair, the expression patterns of DEmiR were opposite to that of DEGs.

### 4.7. Validation of the Identified miRNAs and Targets by Quantitative Real-Time PCR

Twelve total RNAs (four tissue samples with three biological replicates) were extracted from the young central leaves and current year roots of *H. fulva* with and without NaCl treatment and the concentration of total RNA was measured by NanoDrop 2000. For cDNA synthesis, 1 µg of total RNA of each sample was reverse transcribed with miRNA stem-loop primers and oligo dT primers by using the FastKing gDNA Dispelling RT SuperMix (TIANGEN, Beijing, China). Several NaCl-responsive miRNAs and genes including miR396b, miR393a, miR160b, miR156a, and HHC020074 (growth-regulating factor 4-like, GRF4), HHC029515 (transport inhibitor response 1-like, TIR1), HHC035929 (auxin response factor 18-like, and ARF18), HHC018925 (squamosa promoter-binding-like protein 12, SPL12) were selected to validate and quantify using qRT-PCR. Primers were designed according to the miRNA sequences and assembly transcripts using the online Primer-BLAST program (https://www.ncbi.nlm.nih.gov/tools/primer-blast, accessed on 13 October 2021) (Appendix A). The qRT-PCR was performed using the SuperReal PreMix Plus (SYBR Green) (TIANGEN, Beijing, China) on ABI7500 real-time system (Applied Biosystems) according to the manufacturer’s protocol. Three biological replicates and three technical replicates were performed for each of the analyzed genes. The *Hemerocallis Actin* gene was used as an endogenous control for normalization. Relative transcript levels of each gene were calculated with the comparative cycle threshold (2^−ddCt^) method [87].

### 4.8. Statistical Analyses

Statistical data were presented as mean ± standard error (SE). Student’s *t*-test was used to compare the differences between the two groups. Differences were considered to be significant at *p*-value ≤ 0.05. GraphPad Prism v.9.0 (GraphPad Software Inc., La Jolla, CA, USA) was used to perform statistical analysis. 

## 5. Conclusions

In recent years, multi-omics integration analysis has been used to identify plant abiotic stress-responsive miRNAs and target genes. Through small RNA sequencing and transcriptome analysis, we have obtained a great number of differential expression miRNAs and targets in response to salt stress in *H. fulva.* Most conserved salt-responsive miRNAs and target genes identified in the present study have also been elucidated in various plants indicating the common molecular regulatory mechanisms shared by different plant species. Furthermore, miR156, miR160, miR393, miR166, miR396, miR398, and their targets *SPL*, *ARF*, *TIR1*, *CML*, *GRF4*, and *copper/zinc superoxide dismutase*, etc. might be involved in miRNAs and miRNA-mediated regulatory network under salt stress in *H. fulva*. In particular, several novel candidate miRNAs such as PC-5p-361_8563 and PC-3p-4499_652 might participate in salt stress response by targeting *cinnamoyl-CoA reductase* and *hsp70-Hsp90 organizing protein 2* genes in *H. fulva*. Most of the miRNAs and targets are involved in phytohormone signaling, Ca^2+^ signaling, and oxidative defense signaling pathways to regulate NaCl stress response. These results will provide a rich resource for better understanding the molecular mechanism of miRNA/mRNA regulatory net in the NaCl stress tolerance of *H. fulva*. 

## Figures and Tables

**Figure 1 ijms-24-07290-f001:**
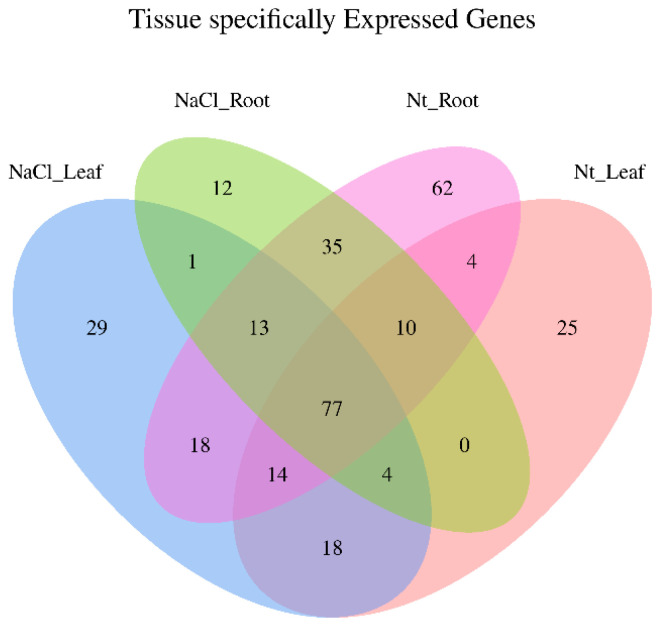
The Venn diagram of miRNAs identified in leaf and root under NaCl stress in *H. fulva*.

**Figure 2 ijms-24-07290-f002:**
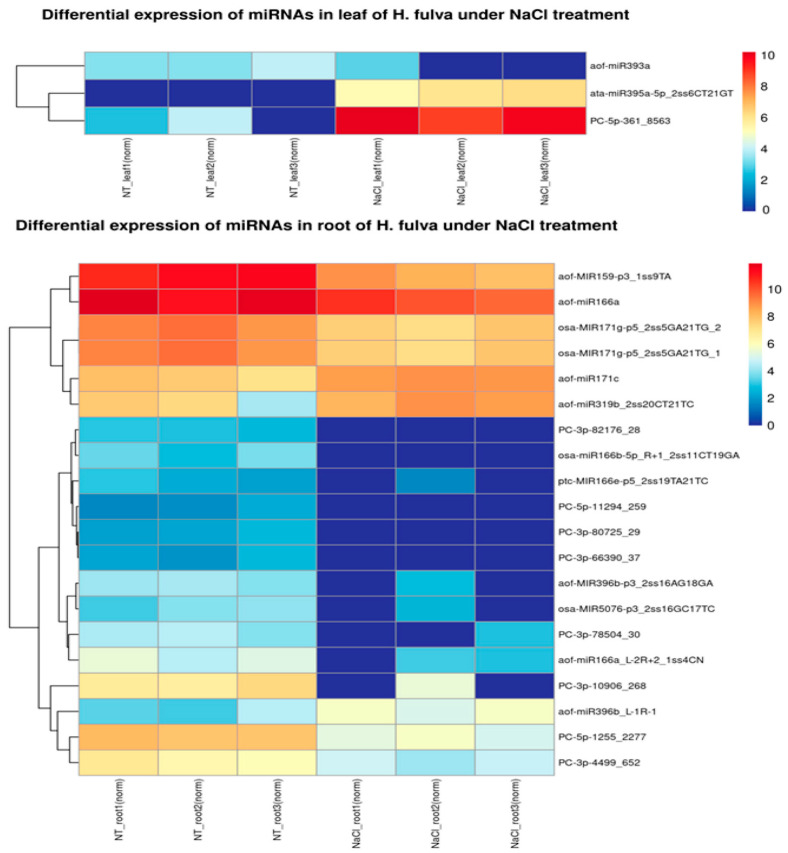
The heatmap of differential miRNAs in leaf and root under NaCl stress in *H. fulva*.

**Figure 3 ijms-24-07290-f003:**
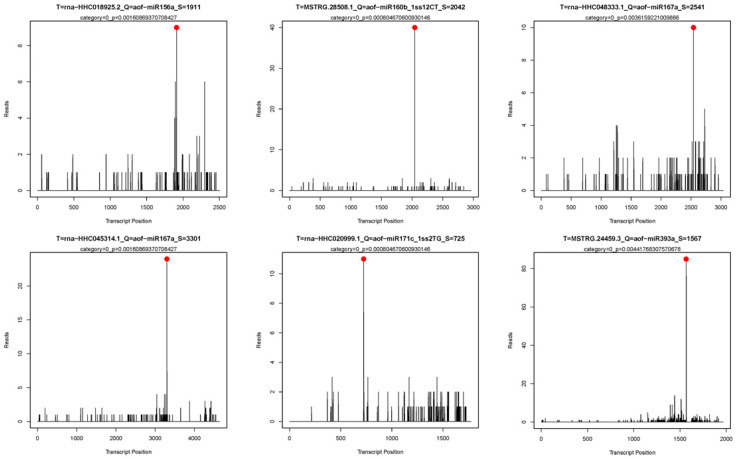
Target plot (t-plots) of candidate miRNA targets confirmed by degradome sequencing. The red lines showed the distribution of the degradome tags along the target mRNA sequences.

**Figure 4 ijms-24-07290-f004:**
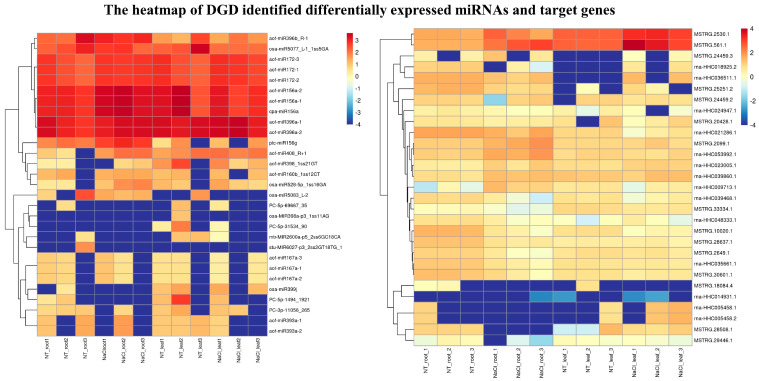
Heatmap of DGD identified DE miRNAs and target genes in leaf and root of *H. fulva* under NaCl stress.

**Figure 5 ijms-24-07290-f005:**
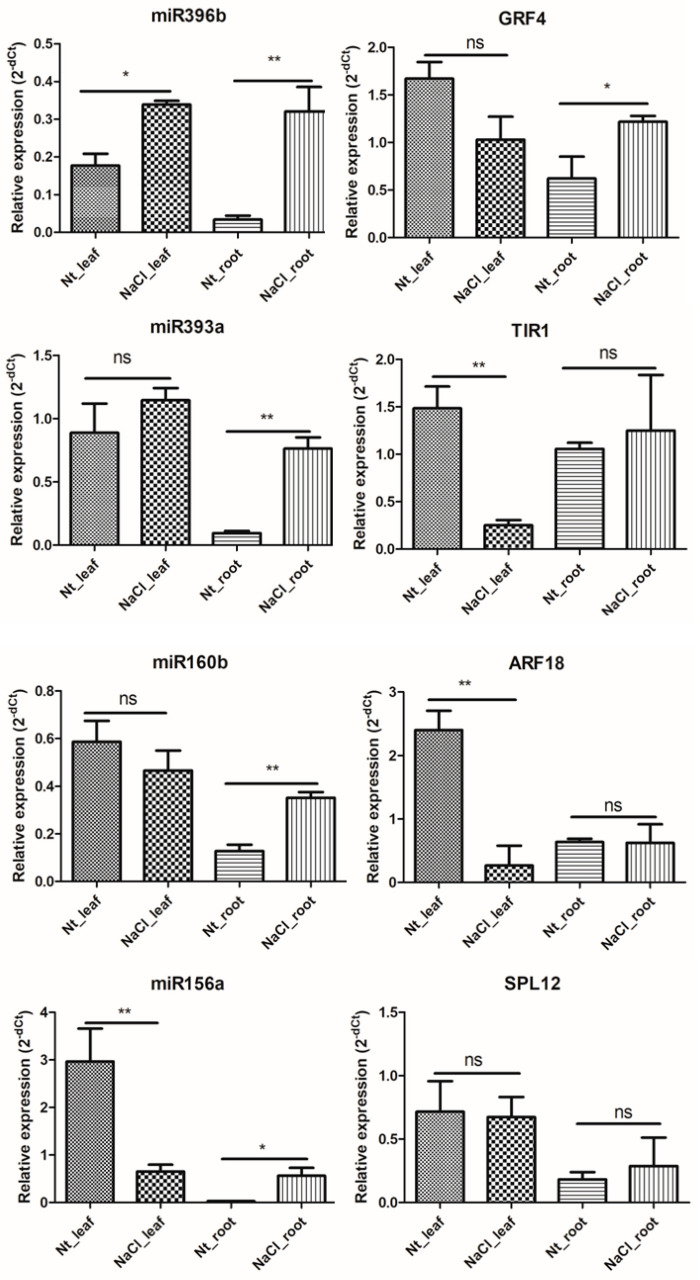
Relative expression analysis of candidate miRNAs and targets in leaf and root of *H. fulva* with and without NaCl treatment by real time PCR. (ns means *p* value > 0.05; * means *p* value ≤ 0.05; ** means *p* value ≤ 0.01).

**Figure 6 ijms-24-07290-f006:**
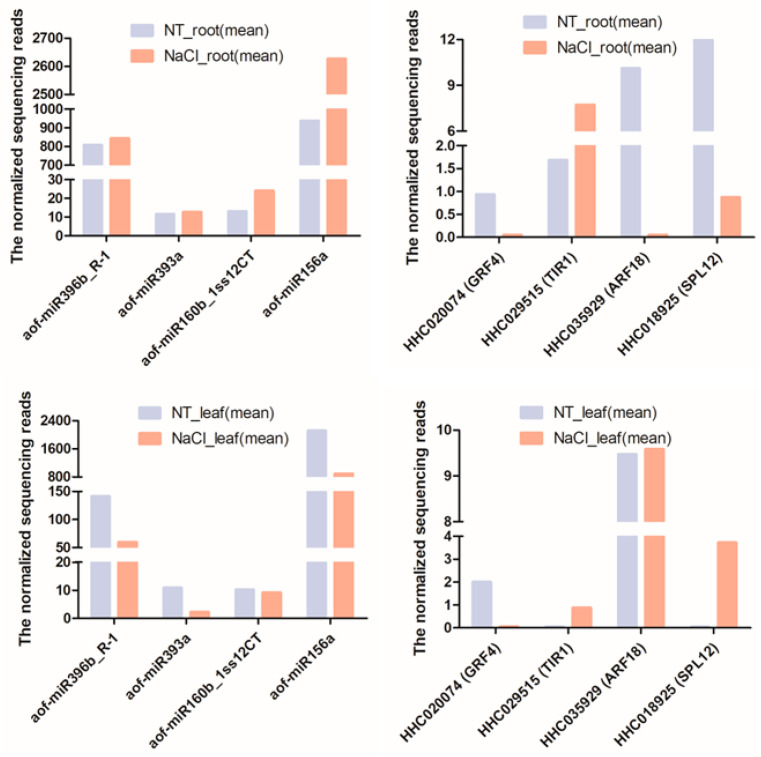
The expression abundance of miRNAs and candidate targets in leaf and root of *H. fulva* identified by high-throughput sequencing.

**Table 1 ijms-24-07290-t001:** Differential expression of miRNAs and their target genes identified in NaCl stress response in *H. fulva*.

miRNA ID	Gene ID	Target Gene	Gene Annotation
aof-miR156a	rna-HHC018925.2	SPL12	squamosa promoter-binding-like protein 12 [Asparagus officinalis]
aof-miR160b_1ss12CT	MSTRG.28508.1	ARF18-like	auxin response factor 18-like isoform X1 [Asparagus officinalis]
aof-miR167a	rna-HHC035661.1	ARF12-like	auxin response factor 12-like isoform X1 [Phoenix dactylifera]
aof-miR167a	rna-HHC048333.1	ARF12-like	auxin response factor 12-like isoform X1 [Phoenix dactylifera]
aof-miR167a	MSTRG.28637.1	ARF12-like	auxin response factor 12-like isoform X1 [Phoenix dactylifera]
aof-miR172	MSTRG.29446.1	A4U43_C07F3620	uncharacterized protein A4U43_C07F3620 [Asparagus officinalis]
aof-miR172	MSTRG.2649.1	A4U43_C07F3620	uncharacterized protein A4U43_C07F3620 [Asparagus officinalis]
aof-miR172	MSTRG.10020.1	A4U43_C07F3620	uncharacterized protein A4U43_C07F3620 [Asparagus officinalis]
aof-miR396a	MSTRG.30601.1	GRF4-like	growth-regulating factor 4-like [Asparagus officinalis]
aof-miR396a	rna-HHC039468.1	GRF4-like	growth-regulating factor 4-like protein [Cinnamomum micranthum f. kanehirae]
aof-miR396b_R-1	MSTRG.18084.4	GRF10-like	PREDICTED: growth-regulating factor 10-like [Elaeis guineensis]
aof-miR398_1ss21GT	rna-HHC005458.2	SOD	copper/zinc superoxide dismutase, partial [Allium sativum]
cpa-miR156a	rna-HHC024947.1	LOC109823482	uncharacterized protein LOC109823482 [Asparagus officinalis]
mtr-MIR2600a-p5_2ss6GC18CA	rna-HHC036511.1	DED37-like	DEAD-box ATP-dependent RNA helicase 37-like [Asparagus officinalis]
osa-MIR398a-p3_1ss11AG	rna-HHC005458.2	SOD	copper/zinc superoxide dismutase, partial [Allium sativum]
osa-miR399j	MSTRG.2099.1	UBE2	probable ubiquitin-conjugating enzyme E2 24 [Asparagus officinalis]
osa-miR5077_L-1_1ss5GA	rna-HHC009713.1	AXF42_Ash001226	hypothetical protein AXF42_Ash001226 [Apostasia shenzhenica]
osa-miR5083_L-2	rna-HHC023005.1	LOC110098070	uncharacterized protein LOC110098070 [Dendrobium catenatum]
osa-miR528-5p_1ss16GA	MSTRG.25251.2	PPO	PREDICTED: polyphenol oxidase, chloroplastic-like [Elaeis guineensis]
PC-3p-11056_265	rna-HHC014931.1	SNAT6	probable sodium-coupled neutral amino acid transporter 6 isoform X1 [Asparagus officinalis]
PC-5p-1494_1921	MSTRG.2530.1	TanjilG_28990	hypothetical protein TanjilG_28990 [Lupinus angustifolius]
PC-5p-69667_35	rna-HHC053992.1	CKAN_02569300	hypothetical protein CKAN_02569300 [Cinnamomum micranthum f. kanehirae]
ptc-miR156g	MSTRG.20428.1	SPL12	squamosa promoter-binding-like protein 12 [Asparagus officinalis]
stu-MIR6027-p3_2ss2GT18TG_1	rna-HHC039860.1	MADS box protein 5	MADS box protein 5, partial [Agave tequilana]
aof-miR408_R+1	rna-HHC021286.1	BCB-like	PREDICTED: blue copper protein-like [Elaeis guineensis]
aof-miR156a	MSTRG.33334.1	A4U43_C04F14680	uncharacterized protein A4U43_C04F14680 [Asparagus officinalis]
aof-miR393a	MSTRG.24459.3	TIR1-like	transport inhibitor response 1-like protein Os04g0395600 [Phoenix dactylifera]
aof-miR393a	MSTRG.24459.2	TIR1-like	transport inhibitor response 1-like protein Os04g0395600 [Phoenix dactylifera]
PC-5p-31534_90	MSTRG.561.1	MT-3a	Metallothionein 3a [Dracaena cambodiana]

## Data Availability

The data are available in Appendix A and the sRNA and transcriptome sequences were deposited in the Sequence Read Archive (SRA) of the National Center for Biotechnology Information (NCBI) (PRJNA951081).

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
