# Peer review of "Integration of mRNA and miRNA Analysis Reveals the Post-Transcriptional Regulation of Salt Stress Response in Hemerocallis fulva"

_ijms, 2023, doi:10.3390/ijms24087290_

Round 1

Reviewer 1 Report

The manuscript describes that identification of differential expression miRNAs and targets in response to salt stress in H. fulva. Especially authors found that several salt-induced related miRNAs participate in salt stress response, including phytohormone signaling, Ca2+ signaling, and oxidative defense signaling pathways to regulate NaCl stress response. Also Authors used RT-PCR confirmed a few special expression miRNA in salt stress condition. So I think it could provide some useful information for readers.

It could be accepted with minor revisions.

1.      Title: Big letter and small letter are not consistency?

2.      Methods:  for RT-PCR, leaf samples from where? In normal condition of stress? Which genes were detected? Please add some data.

3.      Figures could be combined into about 6 Figures. Some of them could be put in supplementary..

4.      The format of reference is not consistency.

5.      English editing is need.  

Author Response

  1. Title: Big letter and small letter are not consistency?

Thank you. We have revised the capital letter of the title.

  1. Methods:  for RT-PCR, leaf samples from where? In normal condition of stress? Which genes were detected? Please add some data.

Thank you for your comments. We have added the source of sample, the condition of stress and the genes of detection in the methods 2.7.

  1. Figures could be combined into about 6 Figures. Some of them could be put in supplementary..

Thank you. We have kept 6 figures and put others in supplementary file.

  1. The format of reference is not consistency.

Thank you for your comments, We have changed the style of references.

  1. English editing is need.  

Thank you for your valuable and thoughtful comments. We have carefully checked and improved the English writing in the revised manuscript.

Reviewer 2 Report

Dear authors! the work was done at a good level. It has a high relevance. my small remarks in the attached file.

Author Response

1. Increase the number of keywords. this will expand interest in the manuscript

Thank you. We have added differentially expressed gene, transcriptome, NaCl treatment in the keywords and written the full title of DEG.

2. A given concentration of salt how detrimental to the plant under study

Thank you. We have added the reference about the concentration of NaCl to treat H. fulva.

Reviewer 3 Report

1. Research questions are well defined and within the aims and the scope of the journal. The introduction is adequate and includes in suitable way the relevant earlier publications. Materials are almost properly described. Methods are also almost properly described and used in a way that is possible to replicate. The investigation is performed to good technical standards. It is no ethical problem involved. A nicely conducted research (although a bit complicated) with conclusions well supported by the results. However, the level of English is inadequate. In some parts the text is not easy to follow. Moreover, the selection of the plant species test needs to be justified! As the manuscript is written it seems of topical interest! Also please use uniform letter fonts.

Therefore, in my opion it can be published prior some revision. Below there are some speciffic concerns.

2. It is necessary to describe the technology of transcriptomics in more detail

3. there is not sufficiant details about  preparing libraries for sequencing, nor anything about the repeats used in the experiment.

4.the authors performed validation by quantitative qRT-PCR? Actually, the information given by the author in the manuscript about the replicas is not clear. According to the standard protocol, it must contain three biological and three technical replicates. Herein there is a lack of this information and thus it seems the qPCR analysis is not properly replicated.

5. there is also no clear information on how many genes have been tested

6. additionally, the full data accompanying these experiments must be made available  in a freely accessible resource.

Author Response

1. Research questions are well defined and within the aims and the scope of the journal. The introduction is adequate and includes in suitable way the relevant earlier publications. Materials are almost properly described. Methods are also almost properly described and used in a way that is possible to replicate. The investigation is performed to good technical standards. It is no ethical problem involved. A nicely conducted research (although a bit complicated) with conclusions well supported by the results. However, the level of English is inadequate. In some parts the text is not easy to follow. Moreover, the selection of the plant species test needs to be justified! As the manuscript is written it seems of topical interest! Also please use uniform letter fonts.

Response: Thank you for your valuable and thoughtful comments. We have carefully checked and improved the English writing in the revised manuscript. The plant species was named ‘Chengsebaoshi’ according to the color of flower which is a cultivated variety of H. fulva. We have added the figure of the species in the supplementary file.

2. Therefore, in my opion it can be published prior some revision. Below there are some speciffic concerns.It is necessary to describe the technology of transcriptomics in more detail

Response: Thank you. We have added the detailed technology of transcriptomics and PCR conditions to obtain cDNA library in the section of 2.2.

3. there is not sufficiant details about  preparing libraries for sequencing, nor anything about the repeats used in the experiment.

Response: Thank you. In the section 2.2, 2.4, the details of preparing libraries for sequencing are added, the data treatment and analysis are also revised.

4.the authors performed validation by quantitative qRT-PCR? Actually, the information given by the author in the manuscript about the replicas is not clear. According to the standard protocol, it must contain three biological and three technical replicates. Herein there is a lack of this information and thus it seems the qPCR analysis is not properly replicated.

Response: Yes, we have performed validation by qRT-PCR. The detailed information have been given in section 2.7.

5.there is also no clear information on how many genes have been tested

Response: Thank you. We have added the name of genes detected with qRT-PCR in section 2.7.

6. additionally, the full data accompanying these experiments must be made available  in a freely accessible resource.

Response: Thank you. The sRNA and transcriptome sequences have been deposited in the Sequence Read Archive (SRA) of the National Center for Biotechnology Information (NCBI) (PRJNA951081).

Round 2

Reviewer 3 Report

I believe that the authors have succesfully adressed my previous concerns, I would only like to draw the authors attention to the following article from which the could use data for discussion.

  • 10.1093/plphys/kiac572

Author Response

 I believe that the authors have succesfully adressed my previous concerns, I would only like to draw the authors attention to the following article from which the could use data for discussion. 10.1093/plphys/kiac572

Response: Thank you for your suggestion. We have cited the reference in the 4.3 of discussion.

4.3. miRNAs regulate NaCl stress response partially via phytohormone signaling, Ca2+ signaling, and oxidative defense signaling pathways

Plants adapt to abiotic stress through miRNA-induced mechanisms at different signaling pathways. Plant hormones play a critical role in plant development and resistance to abiotic stress. In olive (Olea europaea) plants, low-salt pre-exposure primes against high salinity stress, and hormone signaling and defense-related interactions are reported to be associated with the priming responses [70].